# Pleuroparenchymal Fibroelastosis-like Lesions in Clinical Practice: A Rare Entity? Review of a Radiological Database

**DOI:** 10.3390/diagnostics13091627

**Published:** 2023-05-04

**Authors:** Francesco Gentili, Vito Di Martino, Marta Forestieri, Francesco Mazzei, Susanna Guerrini, Elena Bargagli, Antonietta Gerardina Sisinni, Luca Volterrani, Maria Antonietta Mazzei

**Affiliations:** 1Diagnostic Imaging, Azienda Ospedaliera Universitaria Senese, 53100 Siena, Italy; 2Diagnostic Imaging Bambino Gesù Hospital, 00165 Roma, Italy; 3Respiratory Diseases and Lung Transplant Unit, Department of Medical and Surgical Sciences and Neurosciences, University of Siena, 53100 Siena, Italy; 4Unit of Occupational Medicine, Azienda Ospedaliera Universitaria Senese, 53100 Siena, Italy; 5Department of Radiological Sciences, Azienda Ospedaliera Universitaria Senese, 53100 Siena, Italy

**Keywords:** pleuroparenchymal fibroelastosis, computed tomography, lung, interstitial pneumonia

## Abstract

***Background:*** Pleuroparenchymal Fibroelastosis (PPFE) is a rare disease that consists of elastofibrosis that involves the pleura and subpleural lung parenchyma; it is an unusual pulmonary disease with unique clinical, radiological and pathological characteristics. According to recent studies, PPFE may not be a definite disease but a form of chronic lung injury. The aim of this retrospective study is to determine the incidence and to evaluate the distribution, severity and progression of this radiological entity on high-resolution CT (HRCT) exams of the chest, performed in routine clinical practice. In total, 1514 HRCT exams performed in the period January 2016–June 2018 were analyzed. For each exam, the presence of PPFE was evaluated and a quantitative score was assigned (from 0 to 7 points, based on the maximum depth of fibrotic involvement of the parenchyma). When available, two exams with a time interval of at least 6 months were compared for each patient in order to evaluate progression (defined as the increase in the disease score). Patients were divided into different groups according to exposure and their associated diseases. Statistical analysis was performed by using the Wilcoxon test and Kruskal–Wallis test. ***Results:*** PPFE was detected in 174 out of 1514 patients (11.6%), with a mean score of 6.1 ± 3.9 (range 1–14). In 106 out of 174 patients (60.9%), a previous CT scan was available and an evolution of PPFE was detected in 19 of these (11.5%). Among these 19 patients with worsening PPFE, 4 had isolated PPFE that was associated with chronic exposure or connective tissue disorders, and the other 15 had an associated lung disease and/or a chronic exposure. In this group, it was found that the ventral segments of the upper lobes, fissures and apical segments of the lower lobes had a greater statistically significant involvement in the progression of the disease compared to the non-progressive group. In 16 of 174 patients (9.2%, 7 of which belonged to the radiological progression group) a biopsy through video-assisted thoracoscopic surgery or apicoectomy confirmed PPFE. ***Conclusion:*** PPFE-like lesions are not uncommon on HRCT exams in routine clinical practice, and are frequently found in patients with different forms of chronic lung injury. Further studies are necessary to explain why the disease progresses in some cases, while in most, it remains stationary over time.

## 1. Background

Pleuroparenchymal Fibroelastosis (PPFE) is a rare condition that consists of elastofibrosis, involving the pleura and subpleural lung parenchyma with an upper lobe predominance [1,2,3]. The histological pattern of PPFE includes the intense fibrosis of visceral pleura with subpleural intra-alveolar fibrosis, which is associated with alveolar septal elastosis [1,4,5,6]. PPFE is an unusual pulmonary disease with unique clinical, radiological, and pathological characteristics [7].

PPFE was described for the first time in the Japanese literature by Amitani et al., in 1992, as an idiopathic pulmonary upper lobe fibrosis [2]. Frankel et al. identified PPFE as a unique identity that is associated with parenchymal radiographic involvement and an upper lobe predominance; they reported findings during a lung biopsy that did not fit with any of the other defined interstitial pneumonias [3].

In 2013, the American Thoracic Society/European Respiratory Society classified PPFE as being within the idiopathic interstitial pneumonia group [1].

Recent studies have shown the strong association between PPFE and particular clinical states, such us organ transplantation, chemotherapy, infections, dust exposure, connective tissue diseases and vasculitis, suggesting that PPFE may not only be a specific disease, but that it is likely to represent a form of chronic lung injury [8,9,10,11,12].

Moreover, PPFE may coexist with other patterns of interstitial fibrosis in the lower lobes, such as usual interstitial pneumonia (UIP) and nonspecific interstitial pneumonia (NSIP) [6,8]. For these reasons, most authors talk of PPFE-like lesions, given that the histological and radiological characteristics are not exclusive to idiopathic PPFE [13,14].

At present, the incidence and clinical significance of PPFE-like lesions have not been clearly defined, owing to uncertainties in detection and the absence of agreed criteria for the identification of PPFE. A diagnosis of PPFE should ideally be reached after the multidisciplinary consideration of clinical, radiological and, when available, pathological information [7].

The aim of this retrospective study is, therefore, to determine the incidence and to evaluate the distribution, severity and progression of PPFE-like lesions on a radiological database of high-resolution CT (HRCT) exams of the chest, performed in routine clinical practice.

## 2. Methods

This retrospective study was approved by the local Ethics Committee. Cases were identified by reviewing the Diagnostic Imaging Department’s database of the University Hospital of Siena, accounting for examinations performed in the period January 2016–June 2018. Of 1552 CTs initially found, 38 were excluded because the lung was partially included in the scans (n = 13), due to motion artifacts (n = 18) or due to inappropriate technical parameters (n = 7). The remaining 1514 CTs represented the final cohort of our study.

All the CT exams (n = 1552) were performed by using a 64 multi-slice scanner (Discovery CT 750 HD, General Electric Healthcare, Milwuakee, WI, USA). To be eligible, the field of view of each exam had to cover the whole lung parenchyma, from the base to the apex; in total, 832 exams were performed for the evaluation of pulmonary embolism, aortic pathology or oncological pathology, while 720 were performed for HRCT. Since the acquisition parameters changed according to the clinical indication, in this study, only the exams that included a chest scan performed without contrast medium administration, had a layer thickness no larger than 1.25 mm and had been reconstructed with the bone filter, were included. HRCT exams were acquired using the volumetric technique (beam pitch = 0.969: 1; tube rotation = 0.6 s; noise index = 18; kVp = 140; reference mAs = 200–400) and in the prone position in 295 out of the 720 examinations when interstitial lung disease was suspected; this was in order to avoid parenchymal dysventilation in the dependent parts of the lungs, which could simulate fibrotic changes. For each selected patient, we also looked for a previous CT exam (at least 6 months earlier) to examine the progression of the disease and conduct a bioptic investigation; this was in order to obtain a histological confirmation of PPFE.

## 3. Image Analysis

All the CT examinations were analysed in consensus by two radiologists with 25 and 9 years of experience in thoracic imaging, respectively. Readers were asked to review cases, while avoiding any clinical and pathological information. The exams were read through an image reconstruction and interpretation console (Advantage Workstation 4.4, GE Healthcare), correcting the window and magnification values of the image at the discretion of the observer.

The presence of PPFE was recorded according to the radiologic criteria proposed by Reddy et al. in 2012: the presence of pleural thickening with associated subpleural fibrosis in the upper lobes, with a less marked or absent involvement of the lower lobes [5].

For each lung, we evaluated four regions in the attempt to quantify fibrosis.

On the right: apical and/or dorsal segment of right upper lobe (RUL), ventral segment of RUL, apical segment of right lower lobe (RLL) and major fissure.

On the left: apico-dorsal segment of left upper lobe (LUL), ventral segment of LUL, apical segment of left lower lobe (LLL) and major fissure.

For each region, a dichotomous score (0–1) was assigned according to the presence or absence of PPFE alterations. Moreover, we measured the maximum depth of the fibrotic involvement of the parenchyma of both lungs from the pleural plane, regardless of the region analysed; this was performed through multiplanar reconstructions, coronal oblique imaging and imaging that was perpendicular to the pleural plane itself. Then, we assigned a score as follows (Figure 1):0 point: up to 9.9 mm of extension1 point: from 10 mm to 14.9 mm of extension2 points: from 15 mm to 19.9 mm of extension3 points: ≥20 mm of extension

Therefore, the score could vary between 0 (minimum) and 7 (maximum) points for each lung, with a total maximum score of 14 for both lungs. It should be noted that inter-observer agreement was not evaluated for this semi-quantitative assessment. The presence of emphysema in the context of PPFE was assessed visually. Associated pleuro-parenchymal patterns in the remaining portions of the lungs that had not been affected by PPFE were defined radiologically, together with information deriving from other clinical/diagnostic investigations, when available. The same score method was applied to the antecedent CT scans (CTs).

## 4. Statistical Analysis

The analysis was performed by a biomedical statistician.

In selected cases, we investigated the difference in the PPFE score of different categories of patients, as listed in Table 1, using the Wilcoxon test or the Kruskal–Wallis test.

In patients for whom a previous CT scan was available, we calculated the score’s difference between the two different examinations to quantify the progression; in the subgroup of patients that evidenced a difference in their score, we investigated whether this difference was significant by using the Wilcoxon test. Then we compared, in the most recent CT exam, the subgroup of patients with progressive disease to the group with stable disease, regarding the frequency of the involvement of the lung regions listed in Table 2, by using the Wilcoxon test.

*p*-value < 0.05 was considered statistically significant. The scores throughout the results are reported as mean ± standard deviation and range.

## 5. Results

PPFE-like alterations were detected in 174 out of 1514 patients (11.6%, 78 males and 96 females, mean age 63.8 years, range 42–93).

In all patients, an involvement of the apical and/or dorsal segments of the RUL and the apical–dorsal segment of the LUL was found; in 37/174 (21.3%), an involvement of the ventral segment of the RUL was found; in 33/174 (19%), an involvement of the ventral segment of the LUL was found; in 76/174 (19%), an involvement of the apical segment of the RLL was found; in 66/174 (38%). an involvement of the apical segment of the LLL was found; in 99/174 (56.9%), an involvement of the right major fissure was found; and in 96/174 (55.1%), an involvement of the left fissure was found. A score of 3.2 ± 2 (range 1–7) was calculated for the right lung vs. 2.8 ± 2 (range 1–7) for the left lung (*p* = 0.08, no significant difference), for a total mean score of 6.1 ± 3.9 (range 1–14) for each patient. In total, 60 out of 174 (34.5%) patients were smokers or ex-smokers, while the remaining 114 (65.5%) were non-smokers; their scores were 5.3 ± 3.6 (range 2–14) vs. 6.5 ± 4 (range 2–14), respectively (*p* = 0.09, no significant difference).

Regarding the presence of emphysema in the context of PPFE, in 36 out of 174 patients (20.7%), a score of 7 ± 4 (range 2–14) vs. 5.8 ± 3.9 (range 2–14) was calculated in the remaining 138 (79.3%) (*p* = 0.05, no significant difference).

In total, 22 out of 174 (12.6%) patients were affected by oncological diseases treated with chemotherapy, with a score of 5.2 ± 2.9 (range 2–13); 37 out of 174 (21.3%) patients were affected by connective tissue diseases with a score of 4.1 ± 3.7 (range 2–14); 31 out of 174 (17.8%) patients had no known exposure to either associated disease, with an average score of 5.6 ± 3.4 (range 2–14); 51 out of 174 (29.3%) patients had a known asbestos exposure with a score of 4.1 ± 2.9 (range 2–14); and 33 out of 174 (19%) patients had been exposed to other toxic or irritating substances, with a score of 7.4 ± 4.5 (range 2–14). By analyzing multiple comparisons, a significant difference (*p* = 0.002) was only found between the subgroup of patients exposed to asbestos and the subgroup exposed to other substances; no other significant differences (*p* > 0.05) were found.

Considering the pleural diseases and parenchymal diseases associated with the remaining portions of lung parenchyma not involved in PPFE, in 88 out of 174 (50.5%) patients, PPFE lesions were isolated, with a score of 4.9 ± 3.1 (range 2–14); in 12 of 174 (6.9%) patients with emphysema, a score of 6.9 ± 3.7 (range 3–13) was calculated; in 36 of 174 (20.7%) patients with concomitant fibrotic diseases, the score was 9.3 ± 4.3 (range 2–14); in 30 out of 174 (17.2%) patients with concomitant inflammatory/infective bronchial pathology, the score was 5.5 ± 3.8 (range 2–14); and in 8 (4.7%) patients with pleural plaques who had been exposed to asbestos, a score of 3.7 ± 2.2 (range 2–9) was calculated. By analyzing multiple comparisons, a significant difference was found in the PPFE score between patients with bronchial diseases and other pulmonary fibrosis (*p* = 0.002), between other fibrosis and no other disease (*p* = 0.0001), and between other fibrosis and pleural plaques (*p* = 0.006). No other significant differences were found. The PPFE scores of the different patient categories are summarized in Table 1.

In 106 out of 174 patients (60.9%), a previous CT scan was available (mean interval 44.1 ± 28 months, range 6–127), and an evolution of PPFE was detected in 19 of these (11.5%). In patients with exhibiting an evolution of PPFE, the score’s difference between the two CT exams was significant (*p* = 0.004). The mean increase in the score was 2.8 ± 1.4 (range 1–6), with a mean time interval of 54 ± 34.4 months (range 14–118).

In 87 out of 106 patients without progressive disease, apart the involvement of the apical and or dorsal segments of the upper lobes, 39 exhibited an involvement of the apical segment of the RLL, 53 saw an involvement of the right fissure, 15 saw an involvement of the ventral segment of the RUL, 30 saw an involvement of the apical segment of the LLL, 52 saw an involvement of the left fissure, and 12 saw an involvement of the ventral segment of the LUL. On the other hand, regarding the 19 patients with an increased score between the first and the second exam, 17 exhibited an involvement of the apical segments of the lower lobes and fissures bilaterally, while 14 exhibited an involvement of the ventral segments of the superior lobes. Statistical analysis revealed a significant difference between the two groups (progressive vs. non-progressive disease) regarding the involvement of the ventral segments of the upper lobes (*p* = 8.09 × 10^−7^ and *p* = 5.71 × 10^−8^ for the RUL and the LUL, respectively), fissures (*p* = 0.02 and *p* = 0.01 for the right and left fissures, respectively) and apical segments of the lower lobes (*p* = 1.67 × 10^−5^ and *p* = 0.01 for the LLL and RLL, respectively), as shown in Table 2. Among these 19 patients exhibiting a worsening of PPFE-like alterations, 5 showed an associated fibrotic pattern that was “UIP consistent”, 1 of whom worked in a knitwear factory and 1 of whom was exposed to paints. In total, 4 patients showed only PPFE-like lesions, 1 of whom was affected by psoriatic arthritis, 1 who was exposed to asbestos and 2 who were affected by mixed connective tissue disease. In total, 3 patients showed bronchiectasis with infective chronic bronchiolitis, 2 of whom were detected as being positive at BAL for atypical mycobacteria (Figure 2). In total, 2 patients were affected by chronic extrinsic allergic alveolitis, one of whom was a hairdresser and the other a breeder of canaries. Two patients showed a concomitant emphysema.

There was 1 patient who showed a “UIP possible”-associated pattern that worked as a carpenter. In addition, 1 patient had a pleural and parenchymal asbestosis-associated pattern and another had a “UIP inconsistent” pattern of fibrosis. In 16 of 174 patients (9.2%, 7 of which belonged to the radiological progression group), a biopsy was performed through video-assisted thoracoscopic surgery (VATS) or apicoectomy, and the histopathology was compatible with PPFE.

## 6. Discussion

PPFE is a fibrotic disease with unknown etiology and unclear diagnostic criteria, which was classified as a rare interstitial pneumonia in 2013 [1] and has been described in several case reports in the worldwide literature. Although initially it was considered to be a specific and rare idiopathic pathology, the recent literature has shown that it may be considered an epiphenomenon of chronic lung injury in association with various diseases and medical or occupational exposure [9,15]. In this regard, in 2014, Nakatani et al. reviewed their institutional pathology database of surgical lung biopsies, performed from 2004 to 2012 for diagnosing diffuse lung diseases, through the keywords “atelectasis fibrosis” and “pleuroparenchymal fibroelastosis”; they found 14 cases among 205 patients, 2 of which were successively excluded because they exhibited a radiological pattern that was inconsistent with PPFE. In total, 11 of these patients also had a coexisting interstitial fibrotic pattern in the lower lobes: UIP, NSIP or undefined. The authors detected a progression of pleural and parenchymal thickening in 70% of patients in a mean follow up period of 22.5 months [8]. Later, in 2015, Rosenbaum et al. reported 5 cases of histologically documented PPFE, 2 of which were in patients who had been exposed to specific medical treatments (daptomycin and dapsone), and 1 case in which a patient was affected by eosinophilic pneumonia. It is interesting to note that in all cases, fibrosis was not limited to the upper lobes but also spread to the lower lobes [9]. In 2017, Enomoto et al. analyzed a selected population of patients (n = 113) suffering from interstitial lung disease associated with connective tissue disorders; they detected radiological alterations that were consistent with PPFE or similar (PPFE-like, apical cap) in 21 of these, 13 of which exhibited radiological progression [14].

Our study highlights interesting epidemiological data; 174 cases of the 1514 examined had radiological characteristics of PPFE, with a prevalence of 11.5%, 16 (9.2%) of which had been histologically confirmed; a high percentage of patients were exposed to toxic/irritant substances, supporting the etiological hypothesis of chronic pulmonary damage [5,8]. The low percentage of PPFE histological confirmation in our series is due to the non-feasibility of lung biopsy in patients without radiological progression and related symptoms.

Although the high prevalence of PPFE in our case study may seem unrealistic, it should be underlined that, at present, it is not possible to ascertain a distinction between PPFE and the so-called apical cap or fibroelastotic scar when the lesions are mild by using radiological and histological criteria; in fact, both pathological conditions share the fibrous thickening of visceral pleura and the homogeneous and dense intra-alveolar fibrosis with septal elastosis, clearly demarcated from the normal pulmonary parenchyma, with absent or poor granulomas and signs of chronic inflammation [14,16,17,18].

Although apical cap is a frequent condition, is described as apical or biapical fibrosis, has been recognized since the dawn of anatomical pathology and is considered a consequence of pulmonary tuberculosis, various studies have not confirmed the real evidence of granulomas in these scars, discrediting this hypothesis [19,20].

In the 1930s, it was hypothesized that apical cap could be a consequence of the inhalation of silica powders or of low-grade inflammation alongside the secondary collapse of the lung parenchyma due to the poor respiratory movements in that portion [19]. Finally, in 1970, Butler and Kleinerman hypothesized that its etiopathogenesis was a consequence of the hindered healing of chronic infections secondary to the poor physiological perfusion of the apices with consequent repetitive ischemic damage [21].

In our study, it is interesting to underline the apical segments of the lower lobes and fissures’ high percentage of involvement in PPFE-like lesions, particularly in patients without radiological progression; these are elements that, therefore, would not seem useful in distinguishing PPFE from the pleural cap, as supported by some studies [22]. On the other hand, we can note the ventral segments’ high percentage of involvement in patients with radiological progression compared to those with stable disease over time. Regarding patients with radiological progression, it should be noted that in 9 out of 19 cases, an associated pattern of fibrosis was present; however, currently, it is difficult to establish whether PPFE-like lesions represent an epiphenomenon of the underlying diffuse lung disease or are a distinct associated disease.

Independent of this, the UIP pattern has been demonstrated to be a risk factor for a reduced survival time in PPFE [23]. In the other 8 cases of lung infection, a toxic exposure or a connective tissue disease was found; nonetheless, the number of patients with these characteristics and radiological progression represent only a minor percentage of our case study, suggesting that other factors, such as genetic and immunologic mechanisms, may contribute to the development of these changes.

Some limitations of our study should be noted. First, this is a retrospective and monocentric study; second, HRCT findings have not been correlated to lung function tests and to the clinical setting; and third, histological diagnosis was available only in a minority of patients. Finally, it is probable that a certain number of cases do not represent PPFE disease in a strict sense, but instead represent an unspecific apical cap/fibroelastotic scar of a secondary nature that is not idiopathic; this may explain the fairly high incidence of disease in our population.

## 7. Conclusions

PPFE-like lesions are not uncommon on HRCT exams in routine clinical practice. Although some findings in the lung may predict its radiological progression, further studies are necessary to explain why the disease progresses in some cases, but in most remains stationary over time.

## Figures and Tables

**Figure 1 diagnostics-13-01627-f001:**
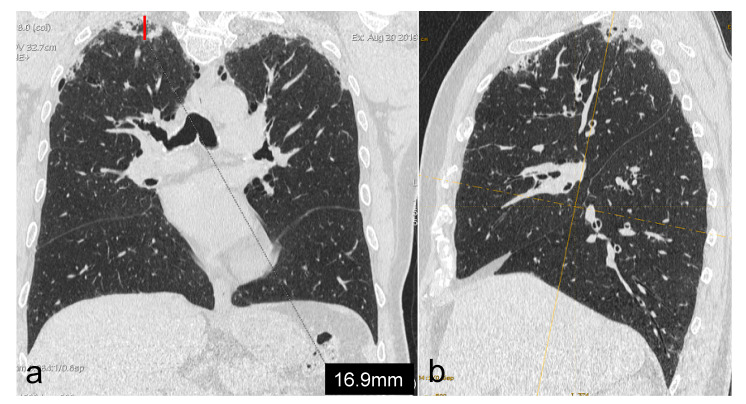
Measurement (**a**) of maximum extension of PPFE-like lesions from the pleural plane into lung parenchyma, by using multiplanar reconstructions, coronal oblique and imaging perpendicular to the pleural plane (**b**).

**Figure 2 diagnostics-13-01627-f002:**
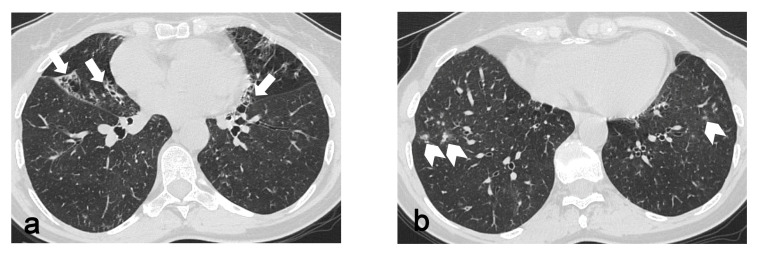
Bronchiectasis (arrows in (**a**)) and infective bronchiolitis (arrowheads in (**b**)) in a 67-year-old woman affected by atypical mycobacteriosis and PPFE-like lesions, which involve upper segments of lower lobes, fissures and apico-dorsal portions of upper lobes (**c**,**d**). A follow-up exam after three years (**e**) shows a progression of PPFE-like lesions.

**Table 1 diagnostics-13-01627-t001:** Summary of the scores of PPFE-like lesions according to the subdivision of patients into various categories on the basis of different characteristics.

Categories	Score (Mean ± SD)	Score (Range)
Smoke		
Smokers/ex-smokers	5.3 ± 3.6	2–14
Non-smokers	6.5 ± 4	2–14
Emphysema in the context of PPFE-like alterations		
Present	7 ± 4	2–14
Aabsent	5.8 ± 3.9	2–14
Associated diseases or exposure		
Oncologic patients	5.2 ± 2.9	2–13
Connective tissue diseases	4.1 ± 3.7	2–14
Asbestos	4.1 ± 2.9 *	2–14
Other toxic substances	7.4 ± 4.5 *	2–14
Any known	5.6 ± 3.4	2–14
Associated pleural or parenchymal diseases in the portions of lung spared by PPFE		
None	4.9 ± 3.1 ^#^	2–14
Other fibrotic patterns	9.3 ± 4.3 *^,#,/^	2–14
Bronchial inflammatory/infective disease	5.5 ± 3.8 *	2–14
Eemphysema	6.9 ± 3.7	3–13
Pleural plaques	3.7 ± 2.2 ^/^	2–9

*^,#,/^ *p* < 0.05 in multiple comparisons.

**Table 2 diagnostics-13-01627-t002:** Summary of the portions of the lung affected by PPFE-like alterations in patients with and without radiological progression.

	No Change (n = 87)	Radiological Progression (n = 19)	*p* Value
Sex (male/female)	31/56	8/11	*p* > 0.05
Apical/dorsal segments of upper lobes	87/87	19/19	*p* > 0.05
Apical segment of RLL	39/87	17/19	*p* = 0.01
Right fissure	53/87	17/19	*p* = 0.02
Ventral segment of RUL	15/87	14/19	*p* = 8.09 × 10^−7^
Apical segment of LLL	30/87	17/19	*p* = 1.67 × 10^−5^
Left fissure	52/87	17/19	*p* = 0.01
Ventral segment of LUL	12/87	14/19	*p* = 5.71 × 10^−8^

## Data Availability

The datasets used and/or analyzed during the current study are available from the corresponding author on reasonable request.

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
