# Peer review of "Pleuroparenchymal Fibroelastosis-like Lesions in Clinical Practice: A Rare Entity? Review of a Radiological Database"

_diagnostics, 2023, doi:10.3390/diagnostics13091627_

Round 1

Reviewer 1 Report

The paper is well structured and addresses a topic of great interest in radiology and pulmonology. In particular, this study highlights the fact that PPFE-like lesions are not so rare in clinical practice and raises the problem of identifying those that require a management in relation to progression. There are some limitations that have largely been pointed out by the authors themselves. Overall the study is good and addresses an important clinical issue whose resolution will require further and more extensive studies. So I think that this paper deserves to be published after some necessary minor revision.

Find enclosed the file with the corrections and comments highlighted in red.

Author Response

The authors thank the reviewer. Corrections are provided in the text in red color.

Reviewer 2 Report

Dear Authors,

the present article showed incidence of PPFE in 11% of the pts underwent Thoracic CT with different indications. Progression of PPFE was identified in near 10% of the cases and associated with chronic exposure, chronic lung disease or CTD. These findings are interesting, nevertheless the study is limited by the low rate of bioptic confirmation of PPFE (around 10% of all PPFE identified). Furthermore, the definition of PPFE progression is not clear in terms of definitions and statistical methods. 

Specific issues listed down below should also be addressed.

Abstract

What do you mean when you state that inclusion and exclusion criteria are defined?

You should define the PPFE score briefly in methods.

You should define in methods the definition of PPFE progression.

Methods

Statistical analysis should be more concise; in addition how did you manage variables? median and IQR? mean and SD?

How can you assess progression with Wilcoxon test?

Results

Exclusion criteria should be defined in methods.

Results are described in a confusing manner. I suggest to summarize as much as possible in table 1; in addition table 1 headings of column 1 and 2 is the same, therefore is difficult to understand.

Discussion

You should discuss the fact that only in few patients you do have histological confirmation of PPFE, this is a main limitation of the study.

Figure 1

You should zoom the apex, the ruler is not visible.

Author Response

Abstract

What do you mean when you state that inclusion and exclusion criteria are defined?

It was indeed not clear. Of the 1552 HRCT exams initially found, 38 were excluded for the reasons explained in details in Methods. For the sake of brevity, we have eliminated the sentence and changed the previous one as follows “1514 HRCT exams performed in the period January 2016-June 2018 were analyzed.”.

You should define the PPFE score briefly in methods.

We have added a brief definition of the PPFE score in the following sentence: “For each exam the presence of PPFE was evaluated and a quantitative score was assigned (from 0 to 7 points, based on the maximum depth of fibrotic involvement of the parenchyma).”.

You should define in methods the definition of PPFE progression.

We have completed the following sentence by adding the last part: “When available, two exams with a time interval of at least 6 months were compared for each patient in order to evaluate progression (defined as the increase of disease score).”.

Methods

Statistical analysis should be more concise;

We have eliminated the detailed description of the data summarized in table 1, substituting the whole paragraph with the following sentence: “In selected cases, we investigated the difference of the PPFE score in different categories of patients as listed in Table 1, using the Wilcoxon test or the Kruskal-Wallis test.” Analogously, we proceeded for the data in table 2: “Then we compared, in the most recent CT exam, the subgroup of patients with progressive disease with that with stable disease, regarding the frequency of involvement of lung regions listed in Table 2, by using the Wilcoxon test.”.

Results

Exclusion criteria should be defined in methods.

We have moved the following sentence from Results to Methods: “Of 1552 CTs initially found, 38 were excluded because the lung was partially included in the scans (n = 13), due to motion artifacts (n = 18) or inappropriate technical parameters (n = 7). The remaining 1514 CTs represented the final cohort of our study.”.

Results are described in a confusing manner. I suggest to summarize as much as possible in table 1;

in addition table 1 headings of column 1 and 2 is the same, therefore is difficult to understand.

We have changed the heading of column 1 with “Score (Mean ± SD)” and that of column 2 with “Score (range)”.

Discussion

You should discuss the fact that only in few patients you do have histological confirmation of PPFE, this is a main limitation of the study.

The low percentage of PPFE histological confirmation in our series is due to the non feasibility of lung biopsy in patients without radiological progression and related symptoms.

Figure 1

You should zoom the apex, the ruler is not visible.

Figure 1 has been modified as request

Reviewer 3 Report

This is a single centre review of all chest CT scan performed between 2016-2018. The authors studied the incidence of PPFE like lesions. PPFE was found in 11.6% scans. Only 9.2% of patients had biopsy confirmation of the diagnosis. The study is interesting and clearly written however the definition of PPFE seems quite broad and the incidence is quite high. This calls into question whether the definition of PPFE is consistent with the prevailing definition. Furthermore they appear to include a cohort of patients with secondary pleural thickening and indeed with other underlying fibrotic lung diseases. While there may be merit to broadening the definition I think the discussion should be tempered to reflect these points. It would also be useful if they could include even a small cohort form another centre to confirm the incidence.

Author Response

This is a single centre review of all chest CT scan performed between 2016-2018. The authors studied the incidence of PPFE like lesions. PPFE was found in 11.6% scans. Only 9.2% of patients had biopsy confirmation of the diagnosis. The study is interesting and clearly written however the definition of PPFE seems quite broad and the incidence is quite high. This calls into question whether the definition of PPFE is consistent with the prevailing definition. Furthermore they appear to include a cohort of patients with secondary pleural thickening and indeed with other underlying fibrotic lung diseases. While there may be merit to broadening the definition I think the discussion should be tempered to reflect these points. It would also be useful if they could include even a small cohort form another centre to confirm the incidence.

A sentence has been added to discussion. It is possible that some cases that do not show radiological progression do not represent PPFE disease in strict sense but only a reaction of pleura and lung to an exogenous stimulus with development of apical cap/ fibroelastotic scar.

Round 2

Reviewer 2 Report

The Authors have addressed all concerns I raised during the 1st revision.

I have no further comments.

Reviewer 3 Report

The authors have modified their definition acknowledging that some cases may not represent true idiopathic PPFE